# Hemostasis Disturbances in Continuous-Flow Left Ventricular Assist Device (CF-LVAD) Patients—Rationale and Study Design

**DOI:** 10.3390/jcm11133712

**Published:** 2022-06-27

**Authors:** Kuczaj Agnieszka, Hudzik Bartosz, Kaczmarski Jacek, Przybyłowski Piotr

**Affiliations:** 1Department of Cardiac, Vascular and Endovascular Surgery and Transplantology, Faculty of Medical Sciences in Zabrze, Medical University of Silesia, 40-055 Katowice, Poland; piotr.przybylowski67@gmail.com; 2Third Department of Cardiology, Faculty of Medical Sciences in Zabrze, Medical University of Silesia, 40-055 Katowice, Poland; bartekh@mp.com; 3Department of Cardiovascular Disease Prevention in Bytom, Faculty of Public Health in Bytom, Medical University of Silesia, 40-055 Katowice, Poland; 4Haemostasis Laboratory, Silesian Center for Heart Diseases, 41-800 Zabrze, Poland; j.kaczmarski@sccs.com

**Keywords:** left ventricular assist device, pump thrombosis, bleeding event, thromboembolic event

## Abstract

Left ventricular assist devices are a treatment option for end-stage heart failure patients. Despite advancing technologies, bleeding and thromboembolic events strongly decrease the survival and the quality of life of these patients. Little is known about prognostic factors determining these adverse events in this group of patients. Therefore, we plan to investigate 90 consecutive left ventricular assist device (LVAD) patients and study in vitro fibrin clot properties (clot lysis time, clot permeability, fibrin ultrastructure using a scanning electron microscope) and the calibrated automated thrombogram in addition to the von Willebrand factor antigen, fibrinogen, D-dimer, prothrombin time/international normalized ratio (PT/INR), and activated partial thromboplastin time (APTT) to identify prognostic factors of adverse outcomes during the course of therapy. We plan to assess the hemostasis system at four different time points, i.e., before LVAD implantation, 3–4 months after LVAD implantation, 6–12 months after LVAD implantation, and at the end of the study (at 5 years or at the time of the adverse event). Adverse outcomes were defined as bleeding events (bleeding in general or in the following subtypes: severe bleeding, fatal bleeding, gastrointestinal bleeding, intracranial bleeding), thromboembolic events (stroke or transient ischemic attack, pump thrombosis, including thrombosis within the pump or its inflow or outflow conduits, arterial peripheral thromboembolism), and death.

## 1. Introduction

Currently, next to transplantation, left ventricular assist devices (LVAD) are a valid option for end-stage heart failure patients. Long-term mechanical circulatory support devices may be considered a bridge to transplantation or even as destination therapy.

Current permanent mechanical circulatory support devices (Heart Ware and Heart Mate III) consist of a pump, a system controller, and a battery unit. The pump is implanted surgically into the thorax, and it is connected to the controller via a driveline that crosses the patient’s abdominal region. The inflow cannula of the pump receives blood from the left ventricle and then the pump transports the blood to the ascending aorta [1].

This treatment allows heart failure patients to return to everyday activities, previous social roles, or even a professional activity [2]. According to the Interagency Registry for Mechanically Assisted Circulatory Support (INTERMACS registry), more than 35,000 patients worldwide receive this type of heart failure therapy. One-year survival is almost the same as in patients after heart transplantation [3]. However, survival and the quality of life are significantly decreased by adverse events, which are typical of this mode of treatment.

Despite antithrombotic therapy (vitamin K antagonist and antiplatelet drugs), ischemic stroke is more prevalent than hemorrhagic stroke, which is relatively rare. However, when it occurs, the risk of death or disability significantly exceeds the risks associated with an ischemic event. The rates of extracranial bleeding are also more frequent compared to those observed in the entire heart failure patient population. Additionally, pump thrombosis is a life-threatening complication resulting in an urgent need for thrombolysis or a pump exchange. The cumulative incidence of adverse events in patients on continuous-flow LVADs reaches up to 30% per single patient [2,4].

### 1.1. Possible Acquired Clotting Disturbances during LVAD Therapy

Patients are prone to bleeding due to antithrombotic therapy and device hemocompatibility (understood as a response of blood components to the artificial material of the device). Additionally, acquired clotting disturbances may occur [5,6]. 

The loss of pulsatile blood flow and low-level hemolysis related to normal LVAD function lead to platelet activation and impaired endothelial function in this group of patients [7]. Preliminary studies showed that the levels of platelet and endothelium-derived microparticles (vascular endothelial cadherin, E-selectin, platelet endothelial cell adhesion molecule, and CD41 + microparticles) increased during LVAD therapy. Elevated levels of microparticles have been shown to correlate with thromboembolic events in these patients and enhance the procoagulant activity of the plasma defined as increased thrombin formation [8]. 

The fibrin clot is an end-stage product of blood coagulation responsible for clinically relevant thromboembolic complications. The clot is mainly composed of fibrin fibers, other components including erythrocytes, platelets, and white blood cells [9]. Some forms of fibrin clots are particularly resistant to lysis and may be associated with an increased risk of thromboembolic events. The most resistant clots are compact, highly branched networks with thin fibers. Fibrin fibers are mainly aligned in the direction of blood flow. This alignment makes them more resistant to lysis [10].

So far, an altered clot structure has been reported in patients with myocardial infarction, ischemic stroke, stent thrombosis, or venous thromboembolism [11,12,13,14]. Siniarski et al. [11] showed that fibrin clots prepared from peripheral blood of patients with acute myocardial infarction were characterized by an altered structure compared to healthy controls. These clots were denser, less permeable, and more resistant to lysis. It is unclear whether LVAD therapy has an impact on this structure or whether the structure is only dependent on specific features of patients.

A key measure of the clot structure is its permeability. This parameter reflects an average pore size between particular fibrin fibers. Compact and dense fibrin clots have low permeability. The clot permeability is calculated from the amount of buffer driven by pressure flowing through a fibrin gel in a given period of time and is defined by Darcy’s constant (Ks). The method has been previously described and defined [15,16].

### 1.2. Drugs Influencing the Clotting Process

The following are among the drugs influencing the clotting process and clot permeability: antiplatelet drugs (acetylsalicylic acid, platelet P2Y12 receptor inhibitors), GPIIB/IIIA inhibitors, and anticoagulant drugs (vitamin K antagonists, direct oral anticoagulants). Medications with an immunomodulatory potential may also have an influence on fibrin clot properties.

Warfarin with target international normalized ratio (INR) of 2–3 increases clot permeability by 20–50%. Studies have demonstrated that statins, angiotensin convertase inhibitors, and glucose-lowering drugs influence the clot structure (Table 1) [17]. We can only hypothesize that similar actions can be attributed to these treatment modalities in LVAD patients.

### 1.3. Role of Inflammation

It has been shown that reduced clot permeability is related to the degree of inflammation and oxidative stress in patients with myocardial infarction [15,17]. However, its occurrence is also possible in other groups of patients. Conditions such as enhanced inflammatory response and oxidative stress are frequently observed in LVAD patients [18], and this association may be similar.

Thus, it is worth investigating the change in the clot structure and function that occurs in non-pulsatile flow in LVAD patients. It may be suspected that the fibrin clot is less permeable and more resistant to lysis in LVAD patients. We assume that genetic factors may influence thromboembolic events in some patients [19]. Therefore, the comparison of changes occurring in particular patients as a result of LVAD implantation may be valuable to investigate.

### 1.4. Fibrinolytic Capacity of the Plasma

The clot lysis time (CLT) reflects the global fibrinolytic capacity of plasma. This method uses the citrated plasma obtained by whole blood centrifugation and assesses the influence of all activators and inhibitors present in plasma. Clotting activation is achieved by the addition of thrombin and calcium, while fibrinolysis is achieved by the addition of plasminogen activator [20].

In patients with atrial fibrillation, it was demonstrated that prolonged CLT was connected with a previous thromboembolic event and stroke [21]. Prolonged CLT was also observed in female patients with thromboembolism in the course of hormonal contraception [22]. We hypothesize that CLT could also be prolonged in LVAD patients at risk of thromboembolic events.

### 1.5. Bleeding during LVAD Therapy

Bleeding is another complication reported in these patients. In LVAD patients, nonsurgical bleeding seems to be correlated with anticoagulation and antiplatelet drugs. Additionally, studies have shown that this form of treatment leads to changes of the following platelet receptors: GPIbα, P-selectin, and PECAM-1. Furthermore, elevated oxidative stress may lead to the exacerbation of nonsurgical bleeding in these patients [23].

The acquired von Willebrand syndrome is a well-known phenomenon in patients treated with CF-LVAD, which leads to more frequent bleeding incidents. Increased shear stress leads to a mechanical destruction of large von Willebrand multimers [24]. As a result, the altered structure of the von Willebrand factor leads to its cleavage by metalloprotease ADAMTS-13. One observational study showed that platelet-thrombus formation in LVAD patients was significantly impaired due to this condition [25]. The mechanism is similar as in severe aortic stenosis. In severe aortic stenosis, non-pulsatile systemic flow and increased shear stress cause the development of angiodysplasia and the acquired von Willebrand syndrome with gastrointestinal bleeding (Heyde syndrome) [26]. It is possible that the von Willebrand factor binds to platelets due to oxidative stress. In patients with aortic stenosis, the severity of stenosis is correlated with oxidative stress and impaired fibrinolysis [27]. We hypothesize that in LVAD patients, fibrinolysis could be disturbed in a similar mechanism. Studies showed that in patients after LVAD implantation with the acquired von Willebrand syndrome and prior gastrointestinal bleeding, there is a seven-fold risk of thromboembolic complications [28]. 

The calibrated Automated Thrombogram (CAT) is a method potentially reflecting the risk of bleeding or thrombosis. This method assesses the overall capacity of plasma to generate thrombin [29]. Studies in patients after LVAD implantation showed that the thrombin generation potential could predict the risk of hemorrhagic complications [30], which is worth investigation.

### 1.6. Scanning Electron Microscopy in Clot Assessment

Scanning electron microscopy (SEM) may provide clinically relevant information about clot nanostructure and cellular components within thrombi [31]. SEM reflects the structure of fibrin networks. This technique of visualization can determine the fiber diameter, length, orientation, and porosity. The ultrastructure is related to clot properties and clinical findings corresponding to the disease. It would be interesting to investigate differences and potential abnormalities characterizing this group of patients that may have potentially relevant clinical implications. This technique of visualization will provide additional information on the above diagnostic methods.

## 2. Aim of the Planned Study

We hypothesize that the hemostasis (e.g., thrombotic factors, fibrin clot structure, and function in particular) undergo substantial changes during the LVAD therapy. Therefore, we set out to determine time-dependent changes of selected hemostasis parameters in patients with end-stage heart failure treated with LVAD and antithrombotic therapy.

### Study Design

The clinical study is a prospective clinical investigation. The study will be conducted in a single high-volume LVAD implantation center. All patients included in the study will receive guideline-directed medical therapy (GDMT) in accordance with the current European Society of Cardiology (ESC) guidelines on management of patients with heart failure [2]. After implantation, all patients will be treated with warfarin to achieve an INR of 2–3 and antiplatelet drugs (acetylsalicylic acid and/or clopidogrel) according to the local standards of LVAD therapy based on the current long-term mechanical circulatory support consensus [4]. The subjects will be given acetylsalicylic acid as a drug of the first choice. The dose will be doubled if the ASPI-test confirms platelet resistance (if the therapeutic level is not achieved). If platelet resistance to acetylsalicylic acid persists despite this intervention, then the regimen will be switched to clopidogrel 75 mg once daily. Dual antiplatelet therapy (acetylsalicylic acid and clopidogrel) would be given for indications other than LVAD therapy (e.g., stent implantation). The management of antithrombotic therapy (both anticoagulant and antiplatelet) will be analyzed and reported. The antithrombotic regimen is given in Table 2.

## 3. Methods

We aim to enroll 90 consecutive end-stage heart failure patients planned for LVAD implantation. The inclusion criteria will be as follows: age above 18 years, no current pregnancy, end-stage heart failure, and fulfilled criteria for LVAD implantation [2]. All patients will be implanted as bridge to transplantation or bridge to candidacy. The exclusion criteria will be as follows: lack or withdrawal of consent for participation in the study, lack of LVAD implantation and contraindications to LVAD therapy (end-stage renal failure, thrombophilia, active infection, contraindication to long-term anticoagulation), severe ventricular arrhythmias), INTERMACS 1 (as these patients are in critical state prior to implantation and the critical condition might have a significant effect on hemostasis), participation in any other clinical investigation involving mechanical circulatory support or interventional investigation that is likely to confound study results or affect the study outcome.

Table 1 Inclusion and exclusion criteria assessed at the time of consent (enrollment).

### 3.1. Timeline of the Investigation

The follow-up is planned as follows: directly prior to LVAD implantation, 3–4 months after implantation, and 6–12 months after implantation. In patients reaching one year of follow-up, we plan to obtain the blood specimens during regular follow-ups (every 6 months after the first year). 

We plan a 5-year follow-up (in case of no adverse events) or examination during/directly prior to the adverse event. Blood samples will be collected after 3 months following implantation, as other studies showed similar results 7 days and 3 months post implantation [8]. Furthermore, we hypothesize that LVAD implantation in INTERMACS 1–2 patients may promote different effects than in other INTERMACS classes. An approval for carrying out the investigation was granted by the Medical University of Silesia Bioethics Committee (PCN/CBN/0022/KB1/144/21/22). The flowchart of the study is given in Figure 1.

### 3.2. Study Objectives and Rationale for Endpoints

The primary endpoint will be met if a non-surgical adverse event occurs (i.e., >14 days after implantation). Net adverse clinical events (NACE) will combine major adverse cardiac and cerebrovascular events (MACCE) and bleeding complications. MACCE will include ischemic stroke or transient ischemic attack (TIA), peripheral embolism, pump thrombosis (including thrombosis within the pump or its inflow or outflow conduits). Bleeding complications will include major bleeding (bleeding requiring transfusion of at least one blood unit), fatal bleeding, gastrointestinal bleeding, and intracranial bleeding. Additionally, we will include driveline infections and other infections requiring intravenous antibiotic administration or hospitalization as secondary endpoints. The primary objective is to analyze the incidence of efficacy and safety outcomes in patients on LVAD managed with antithrombotic strategy. The secondary objective is to link the clinical events to the corresponding changes of the hemostasis. The third objective, if possible, is to create a model (based on hemostasis changes) that would help predict thrombotic and bleeding events. 

### 3.3. Laboratory and Investigation Methods

We plan to assess fibrinogen and D-dimer plasma concentrations, prothrombin time (PT), activated partial thromboplastin time (APTT), international normalized ratio (INR), CAT, CLT, clot permeability (Ks), and SEM ultrastructure of the in vitro clot at the given time points. Additionally, during the follow-up, we plan to assess platelet function (ASPI and ADP test), von Willebrand factor antigen, factor VIII activity, and antithrombin activity. Samples of citrated venous blood will be collected (S—Monovettes: Citrate 9NC/2.9 mL) (9NC:0.106 mol/L).

#### 3.3.1. Prothrombin Time, Activated Partial Thromboplastin Time, Fibrinogen, D-dimer, Antithrombin, Thrombin Time, Factor VIII, von Willebrand Factor

We plan to use the following reagents for the particular assessment:-STA Neoptimal reagent (Diagnostica Stago), clotting method for PT, reference range 70–120%,-STA Cephascreen reagent (Diagnostica Stago), clotting method for APTT, reference range 24–35 s,-STA Liquid Fib reagent (Diagnostica Stago), clotting method for fibrinogen, reference range 200–400 mg/dL,-STA Liatest D-DI Plus reagent (Diagnostica Stago), immunoturbidimetric method for D-dimer, reference range: 0–0.5 µg/mL FEU,-STA Stachrom AT III reagent (Diagnostica Stago), colorimetric method with the use of chromogenic substrate for antithrombin, reference range: 80–120%,-STA Thrombin (Diagnostica Stago), clotting method for thrombin time, reference range: 14–21 s,-STA Immunodef VIII reagent (Diagnostica Stago), clotting method for factor VIII, reference range: 60–150%,-STA Liatest VWF: Ag reagent (Diagnostica Stago), immunoturbidimetric method, reference range: 50–160%.

#### 3.3.2. Platelet Function Assessment

Platelet function assessment (S—Monovettes: Hirudyn/1.6 mL) will be performed in a Multiplate analyzer (Roche). To assess platelet reaction to acetylsalicylic acid, we plan to apply impedance aggregation (ASPI test, Roche) with the reference range from 745 to 1361 AU × min. For Clopidogrel, we plan to apply impedance aggregation (ADP test, Roche) with the reference range from 534 to 1220 AU × min.

#### 3.3.3. Calibrated Automated Thrombogram

CAT in citrated plasma will be performed to assess the thrombin generation potential. We plan to use Diagnostica Stago kit with the application of standardized reagents according to the manufacturer’s recommendations [32]. Thrombin generation kinetics will be measured in the 96-well plate fluorometer (Ascent Reader, Thermolabsystems OY, Helsinki, Finland) equipped with the 390/460 filter set at a temperature of 37 °C. Briefly, 80 μL of PPP will be diluted with 20 μL of the reagent containing 5 pmol/L recombinant tissue factor (TF), 4 μmol/L phosphatidylserine/phosphatidylcholine/phosphatidylethanolamine vesicles, and 20 μL of FluCa solution (Hepes, pH 7.35, 100 mmol/L CaCl_2_, 60 mg/mL bovine albumin, and 2.5 mmol/l Z-Gly-Gly-Arg-amido methyl coumarin).

#### 3.3.4. Clot Lysis Time

To determine plasma lysis potential, we will use CLT, which will be assessed after clotting of the plasma with the addition of 0.5 human thrombin in the presence of 18 ng/mL rtPA (Boerhinger Ingelheim, Ingelheim, Germany). The time of lysis is defined as the time from clear to maximum turbidity (thrombin generation phase) and to the midpoint in the transition from maximum turbidity to final turbidity (clot lysis) on the turbidity curve [33].

#### 3.3.5. Fibrin Clot Permeability

Fibrin clot permeability will be assessed with the application of a pressure system to precisely assess the volume and mass of the buffer flowing through the clot in the function of time. In total, 1 IU/mL of human thrombin will be used as a coagulation trigger. The permeability coefficient will be calculated using the Ks Formula (1):Ks (×10^−9^ cm^2^) = Q × L × η/t × A × ∆P, (1)

(Q (cm^3^)—flow rate at time t (s), L (cm)—length of the fibrin gel, η (dyne × s/cm^2^)—viscosity of the liquid, A (cm^2^) cross-sectional area, ∆P (dyne/cm^2^) is the differential pressure). The details of the assay preparation were previously described [34].

#### 3.3.6. Scanning Electron Microscopic Ultrastructure

SEM techniques will be used to visualize and quantitatively analyze the fibrin clot net obtained from plasma. The rinsed clot will be fixed using 2.5% buffered glutaric formaldehyde, rinsed, dehydrated, dried at the critical point, and gold sprayed. The samples will be scanned in 6 different areas (Jeol JCM-6000 microscope) and analyzed with the application of appropriate software Image J (Bethesda, MD, USA) [34].

## 4. Summary of the Clinical Study

The summary of the planned study is given in Table 3.

## 5. Statistical Analysis

Categorical variables will be presented as counts and percentages. Continuous variables will be presented as the mean and standard deviation for normally distributed data or median with lower and upper quartiles. The Shapiro–Wilk test will be used to verify the normal distribution of the data. The chi-square test will be utilized to compare categorical variables, whereas the t-test or the Mann–Whitney U test will be applied to compare continuous variables where appropriate. A *p*-value < 0.05 will be considered statistically significant. SAS software, version 9.4 (SAS Institute Inc., Gary, NC, USA) will be used for all calculations.

## 6. Discussion of Unique Study Features

After LVAD implantation, the subjects constitute a group of patients at risk of bleeding and thromboembolic episodes despite anticoagulant and antithrombotic therapy.

Due to changes in blood rheology (non-pulsatile flow, development of the acquired von Willebrand syndrome), mechanical circulatory support may lead to clot alterations. We suppose that in some patients the alterations in clot structure may precede the occurrence of adverse events. The impact of inflammation on clot structure and subsequent thromboembolic complications is also interesting. We hope that this investigation will help to find a group of patients strongly at risk of complications before they occur. Potentially, the results of the study may imply a more individualized treatment strategy, depending on a particular risk of bleeding or thrombosis.

## 7. Summary

Despite antiplatelet and anticoagulant treatment, the subjects after LVAD implantation are a group of patients at high risk of thromboembolic complications. Efforts are also made to avoid severe bleeding, which frequently occurs. Advanced functional and microscopic examinations of in vitro clot structure might help the assessment of the particular risk of these destructive complications. Individualization of treatment may be useful to improve the survival and the quality of life in LVAD patients in the future.

## Figures and Tables

**Figure 1 jcm-11-03712-f001:**
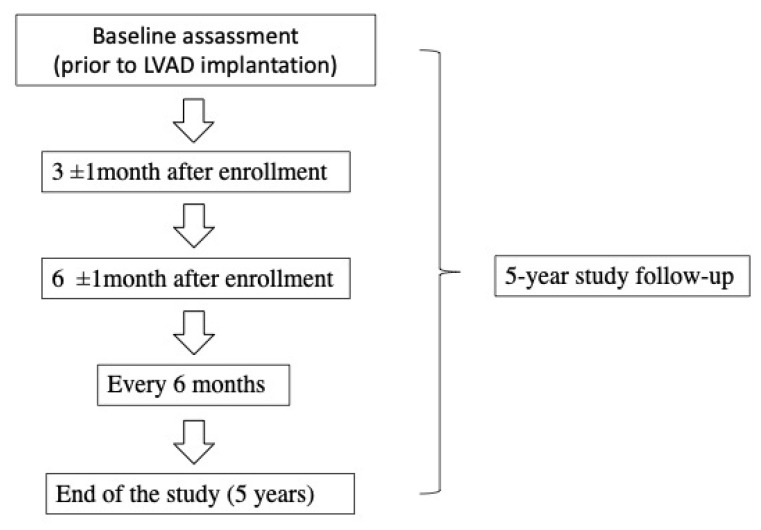
Flowchart of the study. LVAD: Left Ventricular Assist Dev.

**Table 1 jcm-11-03712-t001:** Drugs influencing the clot structure [9,17].

Drug	Mechanism of Action
Metformin, insulin	glycation of fibrinogen and plasminogen, anti-inflammatory properties
Statins	a decrease in tissue factor expression, a reduction of thrombin generation, attenuation of procoagulant reactions catalyzed by thrombin
Acetylsalicylic acid	inhibition of platelet activation, reduced thrombin generation, attenuation of FXIII activation, attenuation of acetylation of fibrinogen, attenuation of the formation of thicker fibrin fibers with larger pores
Warfarin	decrease in thrombin generation
Angiotensin convertase inhibitors	decrease in thrombin generation

**Table 2 jcm-11-03712-t002:** Antithrombotic regimen in LVAD patients.

Drug	Timing	Action	Target
Acetylsalicylic acid	Postoperative day 1 if hemostasis is achieved 75 mg daily	If ASPI test < 745–1361 AU × min, maintain the dose	ASPI test < 745–1361 AU × min
		If ASPI test > 745, uptitrate the dose to 150 mg	
		If ASPI test is still > 745, switch to clopidogrel	
Clopidogrel (instead of acetylsalicylic acid)	In case of acetylsalicylic acid resistance 75 mg daily	If ADP test < 534 AU × min, maintain the dose	ADP test < 534 AU × min
Warfarin	If chest tubes are extracted	Patients bridged with heparin until target INR is achieved	Target INR: 2–3
Unfractionated heparin (if warfarin is temporarily stopped and no heparin-induced thrombocytopenia is found)	As bridge therapy in case of warfarin cessation	Introduce if INR < 2 and maintain until INR > 2	APTT: 60–80 s
Clopidogrel + acetylsalicylic acid	In case of other indications such as neurological, cardiovascular, or hematological	Adhere to separate guidelines	Adhere to separate guidelines

Abbreviations: Adenosine Diphosphate-induced platelet aggregation and secretion (ADP test), arachidonic acid induced aggregation (ASPI-test activated partial thromboplastin time (APTT), international normalized ratio (INR).

**Table 3 jcm-11-03712-t003:** Summary of the clinical study.

Study design	A single-center prospective observational study; enrollment of consecutive 90 pts with end-stage heart failure qualified for LVAD implantation.
Objective	To study hemostatic alterations during LVAD therapy.
Hypothesis	VAD therapy leads to changes in clot structure and function.
Primary endpoint	Survival free from any non-surgical major adverse event at five years. Non-surgical episode is defined as the event occurring >14 days after implantation. Major hemocompatibility-related adverse events: stroke, pump thrombosis, intracranial bleeding, arterial peripheral thromboembolism.
Secondary endpoints	Stroke rates, pump thrombosis rates, bleeding rates, gastrointestinal bleeding. Infection-related adverse events: driveline infections, other infections requiring intravenous antibiotics and/or hospitalizations. Descriptive endpoints: rehospitalization, number of days at hospital.

## Data Availability

Not applicable.

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
