# Peer review of "Hemostasis Disturbances in Continuous-Flow Left Ventricular Assist Device (CF-LVAD) Patients—Rationale and Study Design"

_jcm, 2022, doi:10.3390/jcm11133712_

Round 1
Reviewer 1 Report
The paper represents design of the study, attributed to hemostasis in LVAD patients. Its important question due to high incidence of bleeding and clotting in these patients. The design is well-done.
Author Response
Thank you for your remarks.
The corrections are made in the text (red colored).
Reviewer 2 Report
This manuscript showed a protocol of study to investigate the relationship between bleeding and thrombotic complications and coagulation studies in patients with LVAD. Overall, the research process seems to be well-documented in detail.
1 The ENDPOINT is shown, but it is not clear what the author wants to know based on it. A more detailed description may be needed. Does the author want to know the incidence of bleeding and thrombotic complications in the LVAD patient group at the author's institution? Or are the details of the coagulation tests associated with their occurrence?
This would require a more detailed description in the analysis section. If the author wants to show that patients with thrombotic complications are more prone to thrombosis, or patients with hemorrhagic complications are more prone to bleeding, etc., you will need to do the analysis for that. I think it is necessary to show this in detail.
2 Please write down the approval number of the ethics committee
Author Response

(The authors gave the same response as above.)

Reviewer 3 Report
In their article, Przybyłowski Piotr et al present an extremely interesting signle-centre study design, concerning the potential prognostic role of the hemostatic imbalance in patients undergoing LVAD implantation according to current ESC heart failure guidelines.
The authors explain that balanced hemostasis is the result of complex equilibrium between coagulation and fibrinolysis, and report potential mechanisms responsible for the coagulation imbalance in LVAD patients such as acquired clotting disturbances, drugs, and inflammation. Hence, the aim of this study is clear. The study is meticulously performed in terms of the thorough analysis of the methodology.
The main limitation of this study design is its observational nature and small sample size. The authors should focus on achieving collaborations with other research institutes of their country to reach greater statistical power.
Some minor comments
1) line 53: <<is even 30% per single patient>>. Please make appropriate changes
2)In Table 2 please define the abbreviations << ASPI, ADP>>
3) Section 3. Methods lines 184-185. Please include in the contraindications to LVAD therapy the presentation of severe ventricular arrhythmias.
4)Section 3. Methods, lines 197-198. In my opinion patients with INTERMACS profile 1 are critical and should be considered for LVAD implantation end-organ recovery. Hence, this sub-group is not ideal to be included in your study.
Author Response

(The authors gave the same response as above.)

Reviewer 4 Report
I would just edit to reflect that HeartWare is no longer available for new implants
Interesting study!
Author Response

(The authors gave the same response as above.)

Round 2
Reviewer 2 Report
The manuscript was well revised according to reviewer's comments.
I don't have any further comments for this manuscript.